# Dentine Remineralisation Induced by “Bioactive” Materials through Mineral Deposition: An In Vitro Study

**DOI:** 10.3390/nano14030274

**Published:** 2024-01-27

**Authors:** Marta Kunert, Ireneusz Piwonski, Louis Hardan, Rim Bourgi, Salvatore Sauro, Francesco Inchingolo, Monika Lukomska-Szymanska

**Affiliations:** 1Department of General Dentistry, Medical University of Lodz, 251 Pomorska St., 92-213 Lodz, Poland; marta.kunert@stud.umed.lodz.pl; 2Department of Materials Technology and Chemistry, Faculty of Chemistry, University of Lodz, 163 Pomorska St., 90-236 Lodz, Poland; ireneusz.piwonski@chemia.uni.lodz.pl; 3Department of Restorative Dentistry, School of Dentistry, Saint-Joseph University, Beirut 1107 2180, Lebanon; louis.hardan@usj.edu.lb (L.H.); rim.bourgi@net.usj.edu.lb (R.B.); 4Department of Biomaterials and Bioengineering, INSERM UMR_S 1121, University of Strasbourg, 67000 Strasbourg, France; 5Dental Biomaterials and Minimally Invasive Dentistry, Departamento de Odontología, Facultad de Ciencias de la Salud, Universidad CEU-Cardenal Herrera C/Del Pozo ss/n, Alfara del Patriarca, 46115 Valencia, Spain; 6Department of Interdisciplinary Medicine, University of Bari “Aldo Moro”, 70124 Bari, Italy; francesco.inchingolo@uniba.it

**Keywords:** bioactive dental materials, calcium silicate materials, Ca/P ratio, EDS, pulp capping, MTA, SEM, vital pulp therapy

## Abstract

This study aimed to assess the ability of modern resin-based “bioactive” materials (RBMs) to induce dentine remineralisation via mineral deposition and compare the results to those obtained with calcium silicate cements (CSMs). The following materials were employed for restoration of dentine cavities: CSMs: ProRoot MTA (Dentsply Sirona), MTA Angelus (Angelus), Biodentine (Septodont), and TheraCal LC (Bisco); RBMs: ACTIVA BioACTIVE Base/Liner (Pulpdent), ACTIVA Presto (Pulpdent), and Predicta Bioactive Bulk (Parkell). The evaluation of the mineral deposition was performed through scanning electron microscopy (SEM) and energy-dispersive X-ray spectroscopy (EDX) on the material and dentine surfaces, as well as at the dentine–material interface after immersion in simulated body fluid. Additionally, the Ca/P ratios were also calculated in all the tested groups. The specimens were analysed after setting (baseline) and at 24 h, 7, 14, and 28 days. ProRoot MTA, MTA Angelus, Biodentine, and TheraCal LC showed significant surface precipitation, which filled the gap between the material and the dentine. Conversely, the three RBMs showed only a slight ability to induce mineral precipitation, although none of them was able to remineralise the dentine–material interface. In conclusion, in terms of mineral precipitation, modern “bioactive” RBMs are not as effective as CSMs in inducing dentine remineralisation; these latter represent the only option to induce a possible reparative process at the dentin–material interface.

## 1. Introduction

In modern, minimally invasive restorative dentistry, the preservation of pulp vitality represents one of the main clinical aims. Moreover, according to the European Society of Endodontology (ESE), a more selective caries excavation should be prioritised to complete caries excavation in order to avoid unnecessary pulp exposure, which may lead to the necessity for root canal treatment (RCT) [1,2,3]. There is a growing interest in dental research on innovative materials for the management of pulp using vital pulp therapies (VPTs)—namely pulp capping (PC) [4]. This is because in our modern society, where life span has increased significantly, minimally invasive clinical procedures fulfil the task of postponing non-vital treatments and tooth loss [5]. Indeed, there has been a rapid development of more biocompatible materials with improved “bioactive” properties. Ideally, such materials should be able to stimulate remineralisation of underlying dentine, thereby protecting and preserving pulp vitality. 

In general, the term “bioactivity” refers to the ability of a biomaterial to elicit a specific biological response at the interface between living tissues and the material [6,7]. In restorative dentistry, a material is considered bioactive if it can evoke the precipitation of apatite-like crystals when exposed to body fluids such as saliva [6,8]. The ability of material to release a specific amount of calcium (Ca^2+^) and phosphate (PO_3_^4−^) ions plays an important role in dentine remineralisation via apatite-like deposition [9,10].

Currently, most of the materials indicated as “bioactive” are mainly employed as dental pulp-capping (PC) agents, which are either placed directly on the exposed pulp (direct pulp capping; DPC) or as a cavity liner placed over hypomineralised carious dentine (indirect pulp capping; IPC). Depending on their composition, the materials investigated in this study can be assigned to four clinically significant groups: calcium silicate materials (CSMs), namely ProRoot MTA (Dentsply Sirona, York, PA, USA), MTA Angelus (Angelus, Londrina, Brazil), Biodentine (Septodont, Saint-Maur-des-Fossés, France), and a light-cured calcium silicate-based material TheraCal LC (Bisco, Schaumburg, IL, USA). However, there is a group of resin-based “bioactive” materials (RBMs) in this study constituted by resin-modified glass-ionomer cement (RMGIC) ACTIVA BioACTIVE base liner (Pulpdent Corporation, Watertown, MA, USA), a light-curing composite containing modified calcium phosphate, ACTIVA Presto (Pulpdent Corporation, Watertown, MA, USA), and a dual-cure, bulk-fill resin composite, Predicta Bioactive Bulk (Parkell, Inc., Edgewood, NY, USA) [11,12]. According to the safety data sheets, the only materials recommended for DPC are those belonging to the CSM group; the remaining four resin based materials (TheraCal LC, ACTIVA BioACTIVE, ACTIVA Presto, Predicta Bioactive Bulk) may be suitable only for IPC procedures. ProRoot MTA has been extensively demonstrated to be highly biocompatible and bioactive in dentine remineralisation [13,14]. Due to its main drawbacks, such as the long setting time and high costs, new types of MTA-based materials have been developed [15]. MTA Angelus was introduced with the advantage of a significantly reduced final setting time (24–83 min), and placement that can be performed on a single visit [16,17]. The aforementioned bioactivity and sealing ability of MTA are strongly associated with the calcium ions released from such materials, which then reacts with pulp fluid and induce mineral precipitation and dentine “reparation” [18]. Several studies showed that MTA may be able to induce the formation of hydroxyapatite crystals when immersed in a phosphate buffered saline (PBS), as well as the production of amorphous calcium phosphate (ACP) [19,20]. 

To address the difficulties of using MTA in VPT, novel materials have been designed for more predictable clinical application. For instance, Biodentine (BD), was marketed as a permanent, biocompatible dentine substitute, either facilitating application in one session, or immediately followed by placement of final restoration [21]. Despite the manufacturer claim of a reduced setting time of 9–12 min, it was proven that it only set completely after 45 min; while an intrinsic maturation process continues for up to 14 days [22,23,24]. The interaction of BD with the dentine leads to a marginal sealing by evoking tertiary dentine production and dentine bridge formation [25,26]. ProRoot MTA and MTA Angelus showed some greater toxicity compared to Biodentine, Activa BioActive, and Predicta Bulk Bioactive. Biodentine is the material with the least toxicity and simultaneously the best properties for both indirect and direct pulp capping [27]. BD has become the most used agent for VPT, although long-term clinical studies are still needed to confirm Biodentine as the gold standard material for VPT [28].

TheraCal LC was introduced as a light-cured, resin-modified calcium silicate-based agent that can be used in a single session; it can achieve immediate bonding to resin-based restorative materials [29,30]. The current literature indicates that TheraCal LC is less efficient than MTA and BD as a pulp capping material; this seems to be mainly due to the lower quality of calcific barrier formation, a higher inflammatory effect, less favourable odontoblastic layer formation, and lower Ca-releasing ability [28,31,32]. Moreover, TheraCal LC demonstrates cytotoxic and genotoxic effects on SC cells, making it inappropriate for biological treatment of pulp diseases. A possible cause for its toxicity is the presence of Bis-GMA monomers in its composition [27].

Bioactive materials which facilitate remineralisation of the tooth structure and seal margins by depositing Ca, P, and fluoride (F) ions blur the lines between restorative and preventive dentistry; these are in high demand nowadays. ACTIVA BioACTIVE base liner is a resin-modified glass ionomer cement (RMGIC) containing a modified calcium-phosphate (MCP), which presents a favourable setting time that is attributed to its dual-setting mechanism. It can also be used for one-visit treatment [33]. According to the manufacturer, the bioactive properties of ACTIVA BioACTIVE are attributed to the presence of MCP, as well as the mechanism by which the material responds to pH cycles and releases/recharges significant amounts of Ca, P, and F ions. According to the available literature, ACTIVA BioACTIVE shows lower fluoride-releasing potential than a traditional RMGIC, but higher release of Ca ions as well as a relatively constant phosphate ion release [34,35].

One of the latest developments in bioactive restorative materials is ACTIVA™ PRONTO™, which is available in the US and EU as ACTIVA™ Presto™. It is a resin-based composite that can be used as a bulk-fill restorative material and it contains the patented molecule, Crysta™, which is a modified calcium phosphate functionalised with methacrylate (MCP). It is a stabilised calcium phosphate that is trapped in a transitional state that enables it to deliver the necessary Ca and P ions to the adjacent tooth structure and induce remineralisation. The manufacturer also claims that the material continuously releases and recharges such ions, and supplements those ions naturally present in the saliva and/or provided by dietary sources. Crysta™ also acts as a precursor for nucleation sites that attract and bind Ca and PO ions in the environment, stimulating a slow and natural process of mineralisation. 

Predicta Bioactive is a resin-based material recently introduced to the market. The safety data sheet indicates that the resin matrix contains a poly(2-hydroxyethyl methacrylate) (Poly-2-HEMA) monomer, and that it is free of Bisphenol A-glycidyl methacrylate (BisGMA)-based compounds [36]. However, there is limited information available about its properties, especially with regard to its interaction with the pulp and dentine. Although available research states that Predicta does not exhibit cytotoxic or genotoxic potential, further investigation is necessary to determine its real biocompatibility, bioactivity, and dentine remineralisation capacity [27]. 

It is worth emphasising that there are no studies that evaluated the incidence of mineral deposition as bioactivity markers for novel bioactive materials such as ACTIVA BioACTIVE, ACTIVA Presto, and Predicta Bioactive Bulk in comparison to gold standard CSMs. Thus, this study is aimed to assess the ability of modern resin-based “bioactive” materials (RBMs) to remineralise dentine via mineral deposition and compare the results to those obtained with calcium silicate cements (CSMs). This aim was accomplished by evaluation of the mineral deposition at the dentine–material interface, as well as on the dentine and material surfaces after immersion in simulated body fluid (DPBS) through scanning electron microscopy (SEM) and energy-dispersive X-ray spectroscopy (EDX). Additionally, the Ca/P ratios were also calculated in all the tested groups. The hypothesis of this study was that RBMs would have similar dentine remineralising ability to that of CSMs when immersed in DPBS. 

## 2. Materials and Methods

### 2.1. Sample Preparation and Incubation in DPBS

Sixty-four dentine slices (2 mm thick) were obtained from extracted caries-free human teeth. Institutional Ethical Committee approval was obtained for this study (RNN/173/23/KE). Prior to extraction, patients were informed about the use of the teeth for research purposes, which was confirmed with their written consents. The dentine slices were prepared using a round bur #16 (bur S6801.016 Komet Gebr. Brasseler GmbH & Co. KG z.H., Lemgo, Germany) to create a cavity of 3 mm in diameter and 1.6 mm deep. Sof-Lex™ (3M Oral Care, St. Paul, MN, USA) discs in coarse, medium, fine, and superfine grits were used to create a standardised smear layer. In order to reduce debris, all specimens were cleaned in an ultrasonic cleaner with deionised water.

The specimens were then divided into 8 experimental groups (Figure 1, Appendix A Table A1) and the prepared cavities were filled with the following materials: ProRoot MTA (Dentsply Sirona, York, PA, USA), MTA Angelus (Angelus, Londrina, Brazil), Biodentine (Septodont, Saint-Maur-des-Fossés, France), TheraCal LC (Bisco, Schaumburg, IL, USA), ACTIVA BioACTIVE base liner (Pulpdent Corporation, Watertown, MA, USA), ACTIVA Presto (Pulpdent Corporation, Watertown, MA, USA), Predicta Bioactive Bulk (Parkell, Inc., Edgewood, NY, USA) (Figure 1). The control dentine (control) was etched for 15 s with 37% orthophosphoric acid to eliminate the smear layer and expose the dentinal tubules. The specimens in the groups ACTIVA Presto and Predicta Bioactive Bulk were etched, and then bonded as per manufacturer instructions, using a universal bonding agent (3M Single Bond Universal, 3M ESPE, Seefeld, Germany) (Appendix A, Table A1). The resin-based materials (TheraCal LC, ACTIVA BioACTIVE, ACTIVA Presto, Predicta Bioactive Bulk) were light-cured using a hand light-curing unit (Elipar™ DeepCure-L, 3M ESPE, Seefeld, Germany), while the CSMs were left undisturbed until final setting (ProRoot MTA, MTA Angelus, Biodentine) according to the manufacturer’s specifications (Appendix A, Table A1). The specimens were placed in Eppendorf tubes containing 2 mL calcium-free DPBS (Dulbecco’s Phosphate Buffered Saline; Lonza, Lonza Walkersville Inc., Walkersville, MD, USA) immediately after reaching the setting time stated by the manufacturer. DPBS is a physiological-like buffered (pH 7.4) Ca- and Mg-free solution with the following composition (mM): K^+^ (4.18), Na^+^ (152.9), Cl^−^ (139.5), PO_4_^3−^ (9.56, sum of H_2_PO_4_) 1.5 mM and HPO_4_^2−^ 8.06 mM). Specimens were stored at 37 °C. The evaluation was performed at various incubation times: after setting, 24 h, 7, 14, and 28 days. The samples were examined without prior washing to avoid any potential removal of the deposited minerals.

### 2.2. SEM Imaging 

All of the 32 specimens were sputter-coated with 20 nm of gold and then examined using a scanning electron microscope (SEM, FEI Nova NanoSEM 450, FEI, Hillsboro, OR, USA), with an accelerating voltage of 5 kV, WD = 4.8 ± 0.2 mm. Representative images were captured on the surfaces of the tested materials and at the material–dentine interface to evaluate the reparative/sealing ability of the tested materials. Moreover, the tested materials were also analysed to assess whether they could release enough ions which could reprecipitate on the adjacent dentine surface. The representatives SEM images for each study group were captured at 200×, 500×, 1000×, 3500×, 5000×, and 5000× magnification for the characterisation of the specific microstructure. The assessment was carried out after 24 h and after prolonged incubation in DPBS (7-, 14-, and 28 days).

### 2.3. EDX Analysis

For elemental analysis, a further 32 specimens were examined with an energy-dispersive spectrometer (EDX, EDAX/AMETEK, Materials Analysis Division, Model Octane Super, Mahwah, NJ, USA), using an accelerating voltage of 20 kV. The specimens were then placed directly onto the SEM stub and examined without any previous coating procedures. The surface calcium-to-phosphorus (Ca/P) ratio was then calculated at 24 h and after 28 days of incubation in DPBS. For a comprehensive evaluation, SEM–EDX analysis was also performed on the material and on the adjacent dentine (Figure 1) after 28 days incubation. The latter analysis was also performed 1 mm away from the dentine–material interface to analyse if the tested materials could release enough ions, which could reprecipitate on the occlusal dentine surface as mineral-like crystals. For each specimen, an average of three independent measurements in different locations were considered. 

### 2.4. Statistical Analysis 

The Shapiro–Wilk test was used to confirm the normality of the data. The Scheffé test (post hoc) was used to compare the repeated measurement of ion changes overtime in the tested materials. All statistical analyses were evaluated with the statistical software package Statistica v. 13.1 (StatSoft, Inc., Tulsa, OK, USA), and statistical significance was considered at *p* < 0.05.

## 3. Results

### 3.1. SEM Imaging

The control dentine (control) presented the clear presence of open dentinal tubules during SEM analysis (Figure 2). The SEM analysis also showed that the control dentine presented (Figure 2B,C) some mineral precipitation after 28 days of incubation in DPBS, which suggests that some sort of remineralisation via mineral precipitation can occur spontaneously in simulated body fluid (Figure 2D).

ProRoot MTA after 7 days of DBPS immersion induced the precipitation of a layer of globular crystallites. From day 14 of incubation, irregular aggregates clustered into globular structures of increased thickness and diameter (Figure 3(A2–A4, B2–B4)) were seen. After 24 h, the gap at the dentine–material interface was evident, while after 7, 14, and 28 days of storage, the gaps were filled and covered by minerals (Figure 3(A2–A4)). After 24 h, the surface of ProRoot MTA exhibited a regular, but non-uniform surface with the presence of no mineral precipitation (Figure 3(B1)). Also in this case, from day 14 of incubation, the surfaces were coated by the aforementioned aggregates (Figure 3(B2–B4)). Moreover, the material surface was not only covered by mineral aggregates, but after 14 days smaller crystals deposited uniformly on the entire surface of the dentine, and not only adjacently to the material–dentine interface (Figure 3(C1–C4)). The SEM images also revealed specific flake-like shape deposits, along with globular structures, which were only seen in the case of this particular MTA (Figure 3(D3)). 

As with ProRoot MTA, MTA Angelus also exhibited the presence of irregular mineral deposition after 7 days of incubation. However, from day 14 of incubation a uniform and dense deposition of globular crystallites was detected. The globular structures—firstly seen after 7 days (Figure 4(B2,C2))—were later covered with finer particles. Mineral coating was seen on the surface of the material as well as on adjacent dentine (Figure 4(B3,C3)). The dentine–material interface was filled and covered with the precipitates (Figure 4(A2–A4)). The material surface itself had larger particles than ProRoot MTA. 

Although the calcium silicate materials promoted the formation of a mineral barrier at the dentine–material interface, the SEM images for Biodentine after 7 days incubation showed the most evident sealing effect (Figure 5(A2)). The presence of globular crystalline structures, similar to those observed in ProRoot MTA and MTA Angelus, were clearly detected on the surface of Biodentine (Figure 5(B2,B3)). Over time, such deposits became progressively thicker and more homogeneous, and the adjacent dentine was also coated with a uniform layer of mineral crystallites (Figure 5(C2–C4)). 

TheraCal LC also exhibited at 24 h an interface characterised by the presence of a gap which after 28 days was covered and sealed by minerals (Figure 6(A1–A4)). Characteristic plate-like precipitants were observed on the surface of the material after 7 days of incubation (Figure 6(B2)). Maturation of such crystallites resulted in a thicker layer of denser and finer particles (Figure 6(B1–B4)). After 28 days, the surface of the adjacent dentine was evenly covered with the sporadic presence of globular structures (Figure 6(C4)). 

Higher magnification images of all calcium silicate-based materials tested in the study are presented in Figure 7. All of them generated spherical surface precipitates coated with a uniform layer of mineral crystallites after the 28 days incubation period. SEM revealed specific flake-like shapes in the ProRoot MTA (Figure 7(A1,B1)) and characteristic plate-like structures in the TheraCal LC group (Figure 7(A4,B4)). 

In contrast to the CSMs, Activa BioActive exhibited less potential for mineral precipitation in the presented study (Figure 8). As in in the cases of the CSMs, a gap between the tooth and material was observed, which was only reduced over time compared to the CSMs tested in this study (Figure 8(A1–A4)). The SEM images showed that ACTIVA induced no evident mineral precipitation neither on the material nor the dentine surface during the entire period of incubation up to 28 days (Figure 8). 

The SEM images of the dentine at the Activa Presto interface revealed a stable adhesive layer with no gap after 24 h (Figure 9(A1)). After 28 days, spontaneous cracks at the resin–dentine interface were identified (Figure 9(A4)). In terms of mineral deposition, no remineralisation was observed at the material’s surface during the 28-day incubation period (Figure 9(B1–B4)). After 28 days, only a few precipitates were observed at the dentine adjacent to the material (Figure 9(C4)).

Predicta Bulk created a uniform and well-sealed interface with dentine after 24 h, which then deteriorated after 7, 14 and 28 days (Figure 10(A1–A4)). The inhomogeneous, but regular material surface, exhibited no signs of mineralisation during incubation. The dentine surface presented only the sporadic presence of irregular mineral precipitants (Figure 10(C1–C4)). 

### 3.2. EDX Analysis

ESEM–EDX elemental analysis of ProRoot MTA showed different surface composition depending on the incubation time. The elemental analysis of the specimens after 24 h immersion revealed prevalently C (Carbon), O (Oxygen), Ca (Calcium), Si (Silicon), and Bi (Bismuth) and slight traces of F (Fluoride), Na (Sodium), Mg (Magnesium), and Al (Aluminium). Elemental analysis performed after 28 days showed higher Ca, Si, O, and P content. 

A Ca/P ratio of 5.33 was obtained during the elemental analysis of the mineral spherulites after 28 days (Figure 11, Table 1) which is very far from the Ca/P ratio of apatite. Because of the insignificant amounts of P in the freshly prepared material, assessment was only made after a 28-day maturation period. Elements that significantly decreased during incubation were as follows: C, O, Na, and Bi, whereas a significant increase was observed for Si, P, and Ca (*p* < 0.05) (Table 1).

MTA Angelus after 24 h incubation showed high Ca and O content and traces of Mg, Si, Cl, and F. Subsequent to 28 days in DPBS, the surface was characterised by a higher presence of Ca (33.51% wt) and P (14.81% wt) with a Ca/P ratio of 2.26. Bismuth was detected after 28 days only (0.21% wt) (Figure 12, Table 1). The most significant alterations in composition were observed in the following: O, F, Al, Si, Ca (decrease) and in C, Na, Mg, Zr, P (increase) (*p* < 0.05) (Table 1).

Biodentine after 24 h immersion revealed high Ca, Si, and O content and the absence of P. Zr was also detected in amounts constant over time. After 28 days, this material exhibited similar proportions of the aforementioned elements with statistically significant increased C and Mg content, but with a lower presence of Ca and Cl (*p* < 0.05) (Figure 13, Table 1). 

TheraCal LC after 24 h incubation in DPBS revealed high Ca, P and O content (Figure 14, Table 2). SEM–EDX analysis performed after 28 days showed similar results to those observed at 24 h, but with statistically significant increases in Ca and Zr but a decrease in C (*p* < 0.05) (Table 1). The Ca/P ratio obtained during the elemental analysis of precipitates showed a value of 1.87 after 24 h and 2.08 after 28 days.

Activa Bioactive Base Liner showed stable surface composition and morphologies during the different incubation periods. It revealed the prevalent presence of O, Al, Si, P, and C. A statistically significant decrease was observed in O and increases in Al, Bi, and Ba (*p* < 0.05) (Table 1). The Ca/P ratio from the elemental analysis performed on the material surface decreased over time in DPBS storage (Ca/P—3.50 after 1 day and 2.45 after 28 days). However, the Ca and P levels were significantly lower than in all the other CSMs (Figure 15, Table 1).

ACTIVA Presto showed an unchanged surface morphology after incubation in DPBS. The EDX and the elemental analysis of the specimen after 24 h immersion revealed C, O, and Si as the main components and the presence of Ba (Barium). The study unveiled no particular dynamic in elemental configuration during the 28-day incubation period except statistically significant increase in the Ba content (*p* < 0.05). The Ca/P ratio remained constant (Figure 16, Table 1).

Predicta Bulk revealed no statistically significant changes in composition during the incubation period. EDX showed a high presence of O and C, and low Si and F content, along with traces of Al, Na, P, Ca, and Mg. The Ca/P ratio from the elemental analysis increased over the incubation period from 1.20 after 1 day to 3.38 after 28 days due to a notable decrease in P and Ca (Figure 17, Table 1).

For a comprehensive comparison of the analysed materials, all of the collected data was summarised in Table 1

For a more detailed evaluation of the impact that the tested materials had on the adjacent dentine, SEM–EDX analysis was performed after 24 h (1 mm away from the interface); it was possible to see the presence of Ca content in all the specimens treated with CSMs, but with lower P content (Table 2). It was interesting to note that there was an increase in the Ca/P ratio compared to the non-CSM agents. 

## 4. Discussion

In the present study, ProRoot MTA, MTA Angelus, Biodentine, and TheraCal LC induced important mineral deposition, while ACTIVA BioACTIVE, ACTIVA Presto, and Predicta Bioactive did not show any strong remineralisation activity. The obtained EDX data does not substantiate the notion that any of the examined materials possessed the capability to initiate apatite-like deposition when immersed in DPBS. Nevertheless, the observed increase in the amount of P serves as evidence for the formation of a phosphate-based precipitant. All the tests performed in this study confirmed the low remineralisation potential of non-CSM materials. Therefore, the use of such materials should be considered inferior to both MTA products and Biodentine, and their use should be limited to indirect pulp treatment or to restorative procedures. Hence, the hypothesis tested in this in vitro study was rejected. Previous in vivo studies conducted using hDPCS showed that multilayered cultures of cells had satisfactory attachment to both ProRoot MTA and Biodentine [37,38,39,40]. 

Indeed, the results of the present study revealed that Biodentine is superior to ProRoot MTA and also MTA Angelus in terms of Ca content within the material composition, which may result in more efficient biological and bioactive effects when in contact with biological fluids. On the other hand, MTA Angelus presents higher P content than ProRoot MTA and Biodentine, as demonstrated in previous studies [39,41]. These results are in accordance with the current data from dentine adjacent to the material (Table 2). Interestingly, the Ca/P ratio of the dentine in contact with the cements after 28 days was higher in the Biodentine group (3.50) than both ProRoot MTA (2.91) and MTA Angelus (2.56). This may indicate an increase in mineral uptake from the storage media. Nonetheless, due to a lack of long-term observational studies, it is challenging to definitively determine which material, MTA or Biodentine, is superior in terms of dentine remineralisation, as well as longevity of treatments. However, if one considers factors such as handling and cost-effectiveness, Biodentine seems to be more favourable than MTA cements [42].

It has been advocated that when calcium silicate-based cements (MTA products and Biodentine) are in contact with physiological fluids that contain phosphates, they may produce apatite-like crystals or its precursors [19,41]. The resulting “apatite coating” is believed to be the foundation of the positive biocompatibility of these materials, as well as playing a crucial role in cell attachment, differentiation, and tissue repair associated with mineralised tissue formation [43]. The hydroxyapatite generated by CSMs plays an important role in creating a tight seal between the dentinal walls and the material, which is crucial for the success of most of the endo-resto treatments [44]. Furthermore, the presence of hydroxyapatite encourages the attachment of hard tissue-forming cells to the material surface through the selective adsorption of fibronectin. This, in turn, facilitates the development of biological hard tissue barriers that serve to deter the infiltration of oral cavity bacteria into the dental pulp [45]. 

The present study revealed that all the tested CSMs generated the precipitation of minerals on the surface of both the material and the dentine, with a spherical appearance and acicular microprojections, containing Ca and P, as their principal elements. These globular crystalline structures were regular but non-homogenous in terms of dimensions and morphology. During the hydration process, amorphous calcium silicate hydrates (CSH) were also formed. In fact, EDX analysis revealed the presence of Ca, Si, and O, which are indicative of CSH [39]. Elemental analysis performed after 28 days also showed higher P content, suggesting the presence of calcium phosphate deposits (Figure 11). Interestingly, TheraCal LC, a light-curable CSM, also produced surface precipitates, but with characteristic plate-like structures (Figure 7). These results are in accordance with a recent paper by Maciel et al. [3]. According to the analysis performed on the surfaces of non-CSM materials (ACTIVA BioACTIVE, ACTIVA Presto, and Predicta Bioactive), they remained relatively unaffected during the 28-day incubation period due to their low ability to induce minerals precipitation. These findings are also in accordance with previous studies [42,46]. It is important to consider that the interface between the tested materials and the dentine may have been affected by artifacts due to specimen preparation. For instance, 100% humidity incubation in DPBS and the vacuum needed for gold-sputtering and SEM examination may have altered the structure of such an interface. However, the evident formation of an interfacial layer in all calcium silicate materials has been widely proven and it confirms the results observed in previous studies [47]. The SEM images of the material–dentine interface obtained in the current study clearly demonstrated that Biodentine was the most effective in gap “repairing” and may promote the best results in terms of dentine sealing. In the case of the tested materials applied in combination with adhesive systems (Activa Presto and Predicta Bulk) the results were quite different compared to those of the CSMs. Indeed, the Activa Presto interface was characterised by a stable adhesive layer with no gap after 24 h (Figure 9(A1)). Only after 28 days were spontaneous cracks within the interface observed, but with no presence of mineral precipitation (Figure 9(A4)). Predicta Bulk had a sound bonding interface after 24 h, which then deteriorated after 7, 14 and 28 days (Figure 10(A1–A4)). This indicates that such an interface is quite prone to degradation over time, when immersed in simulated biological fluids [48]. For all the RBMs tested in this study, including ACTIVA BioACTIVE, ACTIVA Presto, and Predicta Bioactive, the SEM analysis of the material–dentine interface showed no evident mineral precipitation. This result should be taken into consideration in further clinical studies when evaluating the risk of microleakage of modern bioactive materials.

ProRoot MTA and MTA Angelus are known for their high levels of Ca, making them ideal for use in VPT procedures such as pulp capping. Biodentine and TheraCal LC also have good release of calcium ions and suitable mechanical and handling properties [49]. The current study demonstrated that CSMs (ProRoot MTA, MTA Angelus, Biodentine, TheraCal LC) induce precipitation of Ca/P crystals on adjacent dentine. It is well known that the presence of calcium ions can promote stem cells differentiation into odontoblast-like cells and migration; these represent crucial issues involved in pulp healing [50,51]. In the EDX analysis, the concentration of Ca was higher in the CSM groups than in the ACTIVA BioACTIVE, ACTIVA Presto, and Predicta Bioactive groups (Table 2). Those findings are in agreement with a previous study that reported the great release of Ca and P ions from Biodentine, which remained constant during a 14-day incubation period [37]. Moreover, according to the available literature, Biodentine and TheraCal LC have a higher release of free Ca ions and diffusion levels compared to both ProRoot MTA and MTA Angelus [52,53]. In the current study, calcium exhibited statistically significant changes between the initial and final measurements for the following materials: ProRoot MTA (increase), MTA Angelus (decrease), Biodentine (decrease), and TheraCal (increase). CSM materials also provide significantly more alkaline pH (9.5–13) than RBM materials; this may indicate an antibacterial capability and enhanced mineralisation [41,54,55]. The highest levels of phosphorous were detected in TheraCal LC (16.39 after 24 h and 17.00 after 28 days) followed by MTA Angelus (0.14 after 24 h and 14.81 after 28 days), ProRoot MTA (0.14 after 24 h and 5.96 after 28 days), Activa BioActive Base/Liner (1.61 after 24 h and 0.70 after 28 days), Predicta Bulk, Biodentine, and Activa Presto. According to presented study, phosphorus showed statistically significant changes between the incubation times for the following materials: ProRoot MTA (increase) and MTA Angelus (increase). It is worth emphasising that CSMs have no phosphate in their composition; therefore, its presence comes from the DPBS. Such an element may be an indicator of phosphates content in the material; these ions are crucial for the neutralisation of the acidic pH created by bacteria and to bind to Ca ions and form calcium–phosphates complexes [56]. 

It is worth emphasising that the Ca/P ratio may impact both the physical and mechanical properties of the dental materials, including their strength, hardness, and biocompatibility [57]. In general, a Ca/P ratio of approximately 1.5–2.0 is considered to be ideal for maintaining the mineralisation and strength of teeth and bones. This is based on the fact that the hydroxyapatite mineral, which is the main mineral in teeth and bones, has a Ca/P ratio of approximately 1.67 [57]. Therefore, maintaining a minimum ratio within this range can ensure the optimal mineralisation and strength of dental hard tissues. In the present study, the Ca/P ratio from the EDX elemental analysis revealed the highest score for ProRoot MTA (5.33 after 28 days). Generally, the CSM materials obtained higher Ca/P ratios than the non-CSM agents in the following (descending) order: ACTIVA BioACTIVE (2.45 after 28 days), MTA Angelus (2.26 after 28 days), TheraCal LC (2.08 after 28 days), Predicta Bioactive (1.72 after 28 days), and ACTIVA Presto (1.38 after 28 days; which may be the Ca/P ratio of the MCP inside ACTIVA) (Table 1). For Ca/P ratio, statistically significant changes were observed between the initial and final measurements for the following materials: ProRoot MTA, MTA Angelus, and Biodentine (all cases showed a decrease). The results obtained in the current study showed no apatite deposition in any of the tested materials. The first comparable assessment in ProRoot MTA was made after a 28-day maturation period. The absence or trace quantity of P in CSMs is supported by other available studies that suggested how the P content in following incubation periods are sourced from storage media (e.g., DPBS) [39,41,58]. Given the initial absence of P and following measurements after 28 days correlated with P intake from the DPBS solution, the Ca/P ratio assessment might be more reliable in the dentin adjacent to the bioactive restoration as described below. The current study did not explicitly address the calcium-to-phosphorus (Ca/P) ratio of the mineral precipitated exclusively at the interface. However, throughout the course of the study, comparative analysis of the elemental results revealed that deposits covering the entire surface exhibited remarkably similar characteristics.

Although ACTIVA BioACTIVE has recently been shown to present some remineralisation ability compared to compomer and glass-ionomer cements, its bioactivity needs to be considered inferior to CSM in terms of mineral deposition [59,60]. For a full evaluation of the impact that the tested materials may have on the adjacent dentine, SEM–EDX analysis was also performed 1 mm from the dentine–material interface. Higher Ca content in dentine was observed for all those specimens in contact with CSMs and lower P, compared to the non-CSM agents. The Ca/P ratio of the group Biodentine was 3.50, followed by ProRoot MTA (2.91), Activa Presto (2.60), MTA Angelus (2.56), Predicta Bulk (2.41), Theracal LC (2.06), and Activa BioActive Base Liner (2.04). Nevertheless, all the tested materials showed a Ca/P ratio above the aforementioned range of 1.67 characteristic for hydroxyapatite. 

The microstructural EDX analysis is in line with other studies, except for those elements with a low proportion (which may be considered “trace elements”) within the materials’ composition [27,40,61]. However, slight variations in the reported chemical composition determined through EDX/EDS analyses may arise due to differences in the equipment utilised and the measurement protocol. Data reported in the present study were based upon the mean result from the three measurements that were carried out for each specimen. In the CSM materials, there were a few compounds that might serve as a radiopacifiers, namely bismuth oxide (ProRoot MTA, MTA Angelus), zirconium oxide (Biodentine), and in newer generations calcium tungstate, which does not cause tooth structure discoloration. Interestingly, in the MTA Angelus group the expected presence of bismuth from bismuth oxide was not detectable after 24 h, however it did register after 28 days (0.21% wt). Those findings are in accordance with previous studies [62]. Bismuth demonstrated statistically significant alterations in measurements from the 24 h to 28-day incubation period for the following materials: ProRoot MTA (showing a decrease) and Activa BioActive Base Liner (showing an increase). As Biodentine contains zirconium oxide as a radiopacifier, Zr were detected in amounts constant over time. According to the conducted study, zirconium exhibited a statistically significant increase between the initial and final measurements for the following materials: MTA Angelus and TheraCal.

ESEM–EDX analysis of Activa Bioactive Base Liner showed stable surface composition and morphology over a period of 28-day storage in DPBS. The specimen revealed the presence of O, Al, Si, P, and Ca, which can be attributed to the oxides of aluminosilicate glass and other glass fillers [63,64]. The Ca and P content detected in Activa Bioactive Base Liner was significantly smaller than in the calcium silicate materials, but the highest in the non-CSM group. The low amount of phosphorus provides evidence to support the manufacturer’s assertion that the product contains a water-friendly ionic resin with phosphate acid functionality and antibacterial properties, which can also facilitate interactions between resin-glass fillers and the tooth structure. This mechanism can lead to the formation of a strong resin-apatite complex, by replacing hydrogen ions with Ca through the phosphate group [65]. Elemental analysis of Activa Presto revealed the highest Al, Si, and Ba peaks among the tested non-CSM group, but the lowest Ca, P and F content. This is probably the reason why it showed less ability to induce mineral precipitation compared to the other tested CSMs. Barium is not typically known for its biological activity but its use as a radiopacifier is a well-established practice in dentistry. In the conducted study, Activa Presto revealed relatively high Ba content (8.49% after 24 h and a statistically significant increase to 9.83% after 28 days) in comparison to the other study groups. During the study, barium also exhibited a statistically significant increase in the Activa BioActive Base Liner group. The concentration of barium in dental materials requires further investigation to ensure that it meets biocompatibility standards. 

The limitations of the present study are related to the in vitro characteristics of the experiments performed to accomplish the aims of this study. The established objectives were based on available research, which suggest that apatite formation may occur in vivo in the presence of specific phosphate-containing fluids such as the constant circulation of biological fluids and blood at the surgical site [38]. In the presented study, bioactive materials provided a source of Ca and P for the remineralisation process as well as the DPBS solution used as a soaking medium. In a clinical environment, there are several variables that could affect leaching of the tested elements, such as saliva composition, dietary source, and also intake from prophylactic products. In this study, the bioactivity of the materials was investigated as a function of the soaking time in DPBS. The DPBS used in this study may not have completely resembled saliva (i.e., its pH and ionic composition), although such DPBS can simulate certain characteristics of the oral cavity. However, testing the bioactivity of dental materials in phosphate buffer solutions has been reported to be a suitable method to induce the precipitation of apatite deposition [19,38,66]. An alternative to a PBS solution is a simulated body fluid (SBF), although its use for bioactivity testing seems unreliable [67].

To analyse the formation of apatite as an index of bioactivity of the cement, other advanced characterisation techniques that also allow in situ observations of physio-chemical properties and the transition kinetics of the researched materials might be used, namely X-ray diffraction (XRD), rheometric analysis, Fourier transform infra-red (FTIR), and Raman spectroscopies [41,68,69]. The visco-elastic behaviour of dental agents can be monitored with rheometric analysis, while phase transformation kinetics, especially essential in the CSM group, may be observed during X-ray diffraction (XRD). To assess the apatite-forming abilities of the cements, other complementary research may revolve around Ca^2+^-releasing capability or the pH of the storage solution controlling its antibacterial activity and leaching abilities [54,60].

A further limitation of the in vitro study is its inability to replicate the long-term clinical performance of materials after they have been present in the cavity for many years. The in vitro model used here does not account for factors such as long-term effects, the absence of a dentine barrier, the immune response in human tissues, and other factors like the age of the patient, all of which have been demonstrated to be significant in determining the success rate of VPT [70]. 

The bioactive materials tested in this study require further investigation, since their exact compositions are not yet clear, making it difficult to understand their real bioactive properties. Nonetheless, the in vitro characterisation of the tested materials presented in this study can serve as an initial evaluation of their potential biological behaviour. Further studies that compare the in vitro tests with the clinical performance of the materials is highly recommended to expand the possibilities of “smart” dental materials, accelerating the shift from passive materials to active fillings as the widely recognised standard of dental care.

## 5. Conclusions

Within the limitations of the present study, it can be concluded that the investigated bioactive dental materials exhibited different remineralisation potential, as follows:ProRoot MTA, MTA Angelus, Biodentine, and TheraCal LC showed significant surface precipitation. Consequently, these materials also formed an interfacial layer between the material and the dentin, filling the gap with precipitates, and demonstrated a higher concentration of Ca within the material. However, Biodentine exhibited the most evident sealing effect at the interfacial site.Thanks to an evident bioactivity, ProRoot MTA, MTA Angelus, Biodentine, and TheraCal LC may be suitable for remineralisation of caries dentine and pulp capping in vital pulp therapies.ACTIVA BioACTIVE, ACTIVA Presto, and Predicta Bulk exhibited inferior mineral precipitation compared to the CSMs so they should be used only for indirect pulp capping and/or restorative procedures.

## Data Availability

The data presented in this study are available on request from the corresponding author.

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
