# Peer review of "Dentine Remineralisation Induced by “Bioactive” Materials through Mineral Deposition: An In Vitro Study"

_nanomaterials, 2024, doi:10.3390/nano14030274_

Round 1

Reviewer 1 Report

Comments and Suggestions for Authors

This study studied resin-based “bioactive” materials to induce dentine remineralisation via mineral deposition. The study is well conducted and detailed information were given.

The evaluation of the mineral deposition was performed through Scanning Electron Microscope (SEM) and Energy Disperse X- ray Spectroscopy 26 (EDX) on the material and dentine surfaces,. Additionally, the specimens were analysed after setting (baseline) and up 28 days.

I would recommend to accept the paper.

One suggest is to delete the word “modern” in the title which is not meaningful.

Comments on the Quality of English Language

This study studied resin-based “bioactive” materials to induce dentine remineralisation via mineral deposition. The study is well conducted and detailed information were given.

The evaluation of the mineral deposition was performed through Scanning Electron Microscope (SEM) and Energy Disperse X- ray Spectroscopy 26 (EDX) on the material and dentine surfaces,. Additionally, the specimens were analysed after setting (baseline) and up 28 days.

I would recommend to accept the paper.

One suggest is to delete the word “modern” in the title which is not meaningful.

Author Response

The reply to reviewr's comments is in the attached file.

Reviewer 2 Report

Comments and Suggestions for Authors

1. The figures and tables in the manuscript are challenging to comprehend. It appears that they require reorganization.

2. Kindly include a scale bar in the SEM image.

3. Incorporate the in vitro cellular activity experiment and enhance the manuscript. It is indispensable for this study.

4. The manuscript's English language requires correction.

Comments on the Quality of English Language

The manuscript's English language requires correction

Author Response

(The authors gave the same response as above.)

Reviewer 3 Report

Comments and Suggestions for Authors

The manuscript "Dentine remineralisation induced by modern “bioactive” materials through mineral deposition: an in vitro study" presents the reactivity of different materials when incubated in DPBS. Some comments:

1) lines 23-25- in the list of materials I suggest to clearly indicate/separate RBM and CSM.

2) line 118 and 130- please indicate "phosphate ions" or "PO4 3- ions" in the place of "PO ions"

3) line 188- end of paragraph 2.1: please add in the text if materials were washed (and how) after incubation and before analyses .

4) scheme 1- 8 groups and 8 specimens per group means 64 specimens. Why 72?

5) line 195 and 206- please see previous comment: 4x8=32

6) line 222 and following- It should be Figure 2. Please number all figures after it. Also, please pay attention to figures numbers later in the text. It is confusing.

7) Please add visible scale bars in all SEM images of all figures

8) line 330- please explain in the text the meaning of a Ca/P ratio of 5.33. What is the meaning of evaluating the ratio in this context/experimental setting?

9) EDX analyses- Ba is not a biocompatible element and may represent a problem in biomaterials. Please give an explanation of where its high amount comes from in some specimens.

10) lines 409-411- the fact that the Ca/P ratio does not match with that of apatite does not mean that an apatite-like material has not been formed. The increase in P amount is an evidence of the formation of a phosphate-material. Other techniques should be used for evaluating the crystalline phase of the deposited material, such as XRD.

11) lines 419-421- The discussion is not supported by data. EDX measurement is hindered by the underlying supporting material that is definitely rich in calcium.

12) line 449- please clarify the meaning of "not homogeneous": in dimensions, thickness, morphology, discontinuous...

13) lines 514-516- the meaning of this discussion is not clear.

14) For supporting the conclusions, mechanical tests should be performed on the materials. Furthermore, the discussion is mainly based on EDX analysis that is poorly reliable. At least XRD should be performed in order to ascertain the presence of apatite.

Author Response

(The authors gave the same response as above.)

Reviewer 4 Report

Comments and Suggestions for Authors

The manuscript describes the study of the ability of modern resin-based “bioactive” materials (RBM) to induce dentine remineralisation via mineral deposition and comparing the results to those obtained with calcium silicate cements (CSM).

The topic is not novel, as there exists large literature and also evidenced by the many statements “…in accordance with (previous studies)….” The scope is narrow and the methods are limited, to only SEM-EDX. The selection of large diversity of modern materials is good.

The omission of the SEM operating parameters compromises reproducibility.

The whole work lacks scientific rigor:

First, there lacks statistical analysis to verify the “significant increase/decrease”.

Second, the work remains qualitative and speculative, in that the main conclusion drawn upon EDX-determined Ca/P ratio lacks concrete scientific base. Even combined with SEM surface morphology, it is not sufficient to justify “none of the tested materials were able to induce apatite-like deposition” without studying if there exists, or not, the “apatite-like” crystal structure using XRD and/or TEM.

Further, result presentation and overall dissemination need to be improved: besides missing scale bars in all the micrographs, raw dump of tedious EDX graphs and tables makes following the work an exhaustive task.

However, the thorough and objective discussion of the limitation of the study as well as the future perspectives are good.

Please provide references to Line 511-516.

Recommend citing Simulations reveal the role of composition into the atomic-level flexibility of bioactive glass cements Physical Chemistry Chemical Physics 2016 besides ref 65.  

Comments on the Quality of English Language

moderate editing is needed

Author Response

(The authors gave the same response as above.)

Round 2

Reviewer 2 Report

Comments and Suggestions for Authors

The manuscript has been well revised and is accepted for publication.

Author Response

The manuscript has been well revised and is accepted for publication.

Dear Sir or Madam,

I would like to express my sincere gratitude for your valuable time and effort dedicated to reviewing the manuscript. Your insightful comments and constructive feedback have been immensely beneficial in enhancing the quality and clarity of the content.

Thank you once again for your commitment to the peer review process. Your contribution has been instrumental in improving the overall quality of the manuscript, and I am truly grateful for your time and expertise.

Reviewer 3 Report

Comments and Suggestions for Authors

Manuscript "Dentine remineralisation induced by modern “bioactive” materials through mineral deposition: an in vitro study." has been somehow improved after revision, but Authors failed in facing the most important issues.

In fact the manuscript remains too qualitative. The discussion remains based only on SEM-EDX analyses which is approximative and that gives data that are affected by large inaccuracy.

I suggest supporting the thesis offered by the SEM-EDX data through the use of additional material characterization techniques.

Quality of presentation has been improved. Figure 2 and all figures reporting SEM: scalebars are too small and hardly visible. Please write new scalebars on the image panels.

Author Response

Dear Sir or Madam,

We wish to express our sincere appreciation for the meticulous evaluation of our manuscript, as well as for invaluable suggestions. We are deeply grateful for the comprehensive feedback provided, which has significantly contributed to the enhancement of the manuscript's quality. We have diligently addressed all the comments and recommendations received during the review process. These changes have been marked in the manuscript using track changes tool.

Manuscript "Dentine remineralisation induced by modern “bioactive” materials through mineral deposition: an in vitro study." has been somehow improved after revision, but Authors failed in facing the most important issues.

In fact the manuscript remains too qualitative. The discussion remains based only on SEM-EDX analyses which is approximative and that gives data that are affected by large inaccuracy.

R: Thank you for your feedback. We acknowledge your observation. We would kindly inform that statistical analysis has been carried out at the request of the reviewer, and the results have been added to the manuscript. Statistical analysis is now described in Materials and Methods:

Lines 230-235: The Shapiro–Wilk test was used to confirm the normality of the data. Scheffé test (post-hoc) was used to compare repeated measurement of ion changes overtime in tested materials. All statistical analyses were evaluated with the statistical software package Statistica v. 13.1 (StatSoft, Inc., OK, USA), and statistical significance was considered at p < 0.05.

Obtained quantitative results are described in Results and also Discussion:

Lines 365-367: In ProRoot MTA group elements that significantly decreased during incubation were: C, O, Na, Bi, whereas a significant increase was observed for Si, P and Ca (p < 0.05) (Table 2).

Lines 376-378: The most significant alterations in MTA Angelus composition were observed in: O, F, Al, Si, Ca (decrease) and in C, Na, Mg, Zr, P (increase) (p < 0.05) (Table 2).

Lines 383-386: The most significant alterations in Biodentine’s composition were observed in: O, F, Al, Si, Ca (decrease) and in C, Na, Mg, Zr, P (increase) (p < 0.05) (Table 2).

Lines 392-394: SEM-EDX analysis of TheraCal LC performed after 28 days showed similar results to those observed at 24 h, but with statistically significant increase in Ca and Zr but decrease in C (p<0.05) (Table 2).

Lines 402-406: In Activa Bioactive Base Liner group statistically significant decrease were observed in O and increase in Al, Bi, Ba (p < 0.05) (Table 2). The Ca/P ratio from the elemental analysis performed on the material surface decreased over time in DPBS storage (Ca/P – 3.50 after 1 day and 2.45 after 28 days). However, Ca and P levels were significantly lower than in all the other CSMs (Figures 15, Table 2).

Lines 412-416: The EDX and the elemental analysis of the Activa Pronto specimens after 24h immersion revealed C, O, Si as the main components and the presence of Ba (Barium). The study unveiled no particular dynamic in elemental configuration during the 28-day incubation period except statistically significant increase in Ba content (p<0.05). The Ca/P ratio remained constant (Figures 16, Table 2).

Lines 421-425: Predicta Bulk revealed no statistically significant changes in composition during the incubation period. EDX showed a high presence of O, C, and low Si and F content, along with traces of Al, Na, P, Ca, Mg. The Ca/P ratio from the elemental analysis increased over the incubation period from 1.20 after 1 day to 3.38 after 28 days due to a notable decrease of P and Ca (Figure 17, Table 2).

Lines 544-547: In current study calcium exhibited statistically significant changes between the initial and final measurements for the materials: ProRoot MTA (increase), MTA Angelus (decrease), Biodentine (decrease) and TheraCal (increase).

Lines 535-536: According to presented study phosphorus showed statistically significant changes between incubation times for the materials: ProRoot MTA (increase), MTA Angelus (increase).

Lines 574-577: For Ca/P ratio, statistically significant changes were observed between the initial and final measurements for the materials: ProRoot MTA, MTA Angelus, Biodentine (all cases showed a decrease).

Lines 615-621: Bismuth demonstrated statistically significant alterations in measurements from the 24h to 28-day incubation period for the materials: ProRoot MTA (showing a decrease) and Activa BioActive Base Liner (showing an increase). As Biodentine contains zirconium oxide as a radio-opacifier, Zr were detected in amounts constant over time. According to conducted study zirconium exhibited statistically significant increase between the initial and final measurements for the materials: MTA Angelus, TheraCal.

Lines 636-641: Barium is not typically known for its biological activity but its use as a radiopacifier is a well-established practice in dentistry. In the conducted study, Activa Presto revealed relatively high Ba content (8.49% after 24h and statistically significant increase to 9.83% after 28 days) in comparison to other study groups. During the study barium exhibited statistically significant increase also in Activa BioActive Base Liner group.

I suggest supporting the thesis offered by the SEM-EDX data through the use of additional material characterization techniques.

R: We express our gratitude for your thorough examination of our manuscript. The time and effort you invested in providing constructive feedback are greatly appreciated.

We acknowledge your interest in gaining a deeper understanding of the precipitate formed in our study. As you correctly pointed out, our primary focus is on elucidating the changes within the material and the surrounding dentin. At present, we refrain from making conclusive statements about the precise composition of the deposited minerals, as a comprehensive analysis may necessitate additional techniques, such as Raman microscopy.

The limitations of our study are described in Discussion:

Lines 662-672: To analyse the formation of apatite as an index of bioactivity of the cement, other advanced characterisation techniques that also allow in situ observations of physio-chemical properties and the transition kinetics of the researched materials might be used, namely X-ray diffraction (XRD), rheometric analysis, Fourier Transform Infra-Red (FTIR), and Raman spectroscopies [43][70][71]. The viscoelastic behaviour of dental agents can be monitored with rheometric analysis, while phase transformation kinetics, especially essential in CSM group may be observed during X-ray diffraction (XRD). To assess the apatite-forming abilities of the cements, other complementary research may revolve around Ca2+ -releasing capability or the pH of the storage solution controlling its antibacterial activity and leaching abilities [62][56].

Your suggestion to include XRD analysis for identifying the exact nature of the precipitates has been duly considered. Although our current study did not delve into the specific composition of the precipitate, this aspect will be addressed in a separate article in the future. At this juncture, I must inform you that we currently lack the capacity to conduct further analyses on the previously prepared samples.

We understand the importance of the suggested analyses and acknowledge their relevance to the advancement of our study.

Once again, we sincerely thank you for your valuable time and expertise.

Quality of presentation has been improved. Figure 2 and all figures reporting SEM: scalebars are too small and hardly visible. Please write new scalebars on the image panels.

R: Thank you for your insight on that matter. Indeed, we have added more visible scale bars to the SEM images.

Reviewer 4 Report

Comments and Suggestions for Authors

The authors have addressed the majority of the concerns and the manuscript can be accepted for publication with further minor revision on the quality of presentation:

1. Please add visibal scale bars to the SEM micrographs in addition to those in the program-generated output images.

2. 17 figures are too many, eliminate all EDX ones.

Author Response

Dear Sir or Madam,

I would like to express my sincere gratitude for your valuable time and effort dedicated to reviewing the manuscript. Your insightful comments and constructive feedback have been immensely beneficial in enhancing the quality and clarity of the content.

The authors have addressed the majority of the concerns and the manuscript can be accepted for publication with further minor revision on the quality of presentation:

Please add visibal scale bars to the SEM micrographs in addition to those in the program-generated output images.

R: Thank you for your insight on that matter. Indeed, we have added more visible scale bars to the SEM images.

17 figures are too many, eliminate all EDX ones.

R: Thank you for your feedback. Indeed, we have rewritten Materials and Methods section as per your suggestion:

Thank you very much for your thoughtful review of our manuscript. We appreciate your attention to detail. We acknowledge your observation regarding the abundance of figures in the manuscript. However, we have made a deliberate choice to retain the EDX results as they constitute a crucial component of the conducted research and serve as a meaningful complement to the SEM findings. The inclusion of these graphs allows for a visual comparison of the elemental compositions of the materials under investigation.

We genuinely value your suggestion and the time you dedicated to reviewing our work. Your input has been instrumental in refining our manuscript, and we are grateful for your constructive feedback.

Round 3

Reviewer 3 Report

Comments and Suggestions for Authors

The manuscript has been improved since the first version, as well as the quality of presentation.